# SA-Net: A scale-attention network for medical image segmentation

**Jingfei Hu**[1,2,3,4], **Hua Wang**[1,2,3,4], **Jie Wang**[5], **Yunqi Wang**[1,2], **Fang He**[2], **Jicong Zhang**[1,2,3,4,6]*

1 School of Biological Science and Medical Engineering, Beihang University, Beijing, China, 2 Hefei Innovation Research Institute, Beihang University, Hefei, China, 3 Beijing Advanced Innovation Centre for Biomedical Engineering, Beihang University, Beijing, China, 4 School of Biomedical Engineering, Anhui Medical University, Hefei, China, 5 School of Computer Science and Engineering, Beihang University, Beijing, China, 6 Beijing Advanced Innovation Centre for Big Data-Based Precision Medicine, Beihang University, Beijing, China

* jicongzhang@buaa.edu.cn

**Data Availability Statement:** All relevant data are within the manuscript.

**Funding:** This work was supported by the National Key Research and Development Program of China under Grant 2016YFF0201002, the National Natural

## Abstract

Semantic segmentation of medical images provides an important cornerstone for subsequent tasks of image analysis and understanding. With rapid advancements in deep learning methods, conventional U-Net segmentation networks have been applied in many fields. Based on exploratory experiments, features at multiple scales have been found to be of great importance for the segmentation of medical images. In this paper, we propose a scale-attention deep learning network (SA-Net), which extracts features of different scales in a residual module and uses an attention module to enforce the scale-attention capability. SA-Net can better learn the multi-scale features and achieve more accurate segmentation for different medical image. In addition, this work validates the proposed method across multiple datasets. The experiment results show SA-Net achieves excellent performances in the applications of vessel detection in retinal images, lung segmentation, artery/vein(A/V) classification in retinal images and blastocyst segmentation. To facilitate SA-Net utilization by the scientific community, the code implementation will be made publicly available.

## Introduction

Since manual and dense labeling of a large number of medical images is time-consuming, tedious and prone to inter- and intra-observers, automatic methods for medical image segmentation have been rapidly emerging. These methods lead to accurate and reliable solutions that could improve clinical workflow efficiency and support health-care decision making by allowing quick and automatic extraction of useful quantitative information.

Good representation capabilities, efficient inference, and filter sharing make convolutional neural networks (CNN) the *de facto* standard for image segmentation. Full convolutional networks (FCN) [1] demonstrated good semantic segmentation performance on the Pascal VOC dataset. U-Net [2] and U-Net variants models have been successfully used in segmenting biomedical images of neuronal structures. In particular, MultiResUNet [3] achieved the best segmentation accuracy for neuronal structures in electron microscopy. Recently, both FCN and

Science Foundation of China under Grant 61572055, the University Synergy Innovation Program of Anhui Province GXXT-2019-044, Hefei Innovation Research Institute, Beihang University, and 'the Thousand Talents Plan' Workstation between Beihang University and Jiangsu Yuwell Medical Equipment and Supply Co. Ltd. These awards were received by Jicong Zhang. The funders played a role in study design, data collection and analysis, decision to publish, and preparation of the manuscript.

**Competing interests:** This research was supported by Jiangsu Yuwell Medical Equipment and Supply Co. Ltd. This does not alter our adherence to PLOS ONE policies on sharing data and materials. And the authors declare that no competing interests exist.

U-Net have achieved semantic segmentation performance that closely matches the radiologist performance in many tasks [4–10].

In recent years, many U-Net variants have been devised for different tasks of medical image segmentation. Fu *et al.* [4] adopted a conditional random field (CRF) to extract multi-stage features for improving vessel detection outcomes. The M-Net [5] architecture, a variant of U-Net, was proposed for joint segmentation of the optic disc and cup by augmenting U-Net with deep supervision and multi-scale inputs. Alom *et al.* [6] proposed RU-Net, a U-Net variant equipped with recurrent convolution. Ozan *et al.* [7] improved the U-Net performance with an attention mechanism. Simon *et al.* [8] proposed the Tiramisu architecture in which dense blocks convolutions replace the U-Net convolutional layers. Other CNN variants, such as PSPNet [9] and DeepLab [10], were introduced to achieve superior performance on benchmark tasks of semantic segmentation. Despite the emergence of the afore-mentioned variants, U-Net is still the most common architecture for medical image segmentation, essentially as its encoder-decoder organization (together with its skip connections) does not hinder efficient information flow, and its performance does not deteriorate at low data regime.

U-Net and U-Net like models has been showing impressive potential in segmenting medical images, but the performance of these models will be poor when the target organ exhibits large shape and size variations among patients. Therefore, design good multi-scale features for medical images segmentation is essential. However, creating multi-scale representations requires feature extractors to use receptive fields of considerable variations to give a detailed account of parts, objects, or context at all possible scales. The natural way for CNNs to extract coarse-to-fine multi-scale features is to utilize a convolutional operator stack. Such inherent CNN capability of extracting multi-scale features leads to good representations for handling numerous medical image analysis tasks.

To handle the problem of scale variations, Adelson et al. [11] intuitively leveraged multi-scale image pyramids, and a technique that is quite common in approaches based on hand-crafted features [12, 13] and CNN features. There is a concrete evidence [14, 15] that multi-scale feature learning could be beneficial for deep-learning detectors [16, 17]. The sensitive nonlinear iterative peak (SNIP) algorithm [18, 19] achieves scale normalization by selectively picking training objects of suitable dimensions for each image scale. This algorithm avoids objects of extreme scales, i.e., small or large objects under relatively smaller or larger scales, respectively. However, the computationally high inference times of the image pyramid methods make these methods practically infeasible. The CE-Net [20] architecture employs Dense Atrous Convolution (DAC) blocks to create a multi-scale network for better medical image understanding. Atrous/Dilated convolution [10] expands convolutional kernels by carrying on convolution at sparsely sampled positions. Dilated convolution is frequently utilized in semantic segmentation to account for large-scale contextual information [21, 22]. However, it still suffers from some potential shortcomings, for example, it may cause some pixels never participate in the calculation, which is not friendly to pixel-level prediction. In addition, although the dilated convolution guarantees a larger receptive field with no additional parameters, it is extremely unfriendly for some small objects that do not need such a large receptive field. Moreover, in comparison to the conventional FCN, ResNet-101 [23] has 23 residual blocks (with 69 convolutional layers) of Dilated FCN which require 4 times more computational operations and memory resources, while 3 residual blocks (with 9 convolutional layers) need 16 times more resources. Recently, Res2Net [24] has been constructed as individual residual blocks where each block has hierarchical residual connections. Res2Net adopts a granular-level representation of multi-scale features and enlarges the receptive field range for every network layer. Yet, without exploiting information at different scales, many pieces of redundant information are also transmitted to large-scale features.

Motivated by the afore-mentioned approaches, we make the following key contributions:

- We propose a new scale-attention deep learning network (SA-Net) based on the residual module and attention module, which could segment the different medical images effectively.

- To capture more multi-scale features to better comprehend the structure and function of different tissues in a medical image, an effective Scale-Attention (SA) module is introduced.

- The proposed method is tested in lung segmentation, retinal vessel detection, artery/vein (A/V) classification and blastocyst segmentation tasks. The experimental results demonstrate a superior performance on various tasks compared to the competing methods.

The remainder of the paper is structured as follows. We outline the proposed deep learning framework in Section Methods. The experimental settings are described, and the results and discussion are presented in Section Experimental evaluation and results. Our key conclusions are stated in Section Discussion and conclusions.

## Methods

This section presents in detail the design of the scale-attention network for medical image segmentation. Firstly, we use a basic U-Net as the backbone network. In addition, we insert a SA module in the bridge connection of the U-Net, which could extract multi-scale residual features for achieve our aim of a scale-attention network for different tissues segmentation in medical images. Fig 1 outlines the SA-Net framework.

Our main target herein is to obtain a feature map that can be learned to integrate different scale representations according to different tissue scales in the input medical images. For example, in the retinal fundus images (Fig 2), the main blood vessels branch into micro blood vessels, and the blood vessel diameters in the image range from 1 to 35 pixel units. Fig 2 shows that the micro blood vessels are of very high frequency. Therefore, understanding the huge scale variations are quite crucial and challenging.

### Multi-scale features

In this work, we use the capabilities of Res2Net [24] to learn and understand the image features at different scales. Instead of extracting features using a group of 3×3 filters as in the ResNet

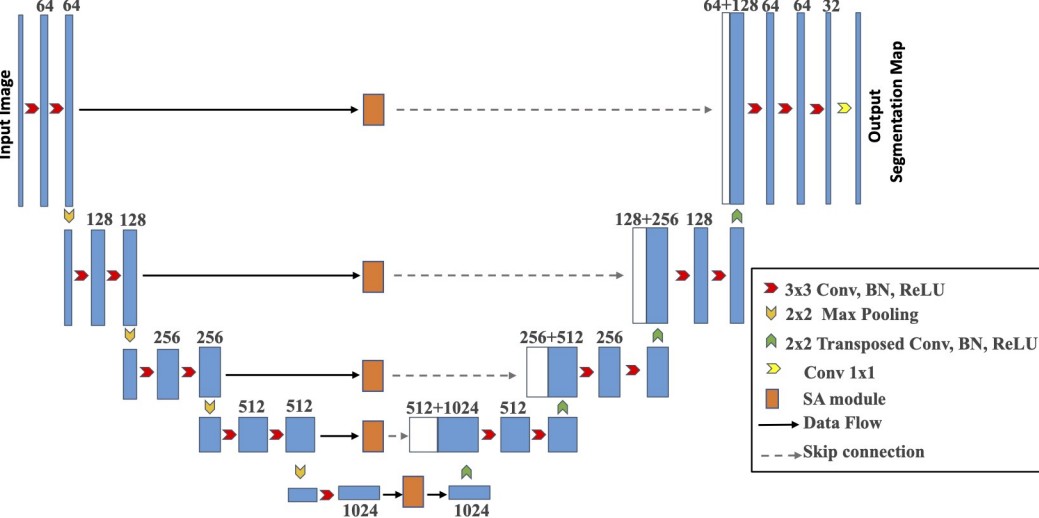

**Fig 1. Diagram of the proposed SA-Net.**

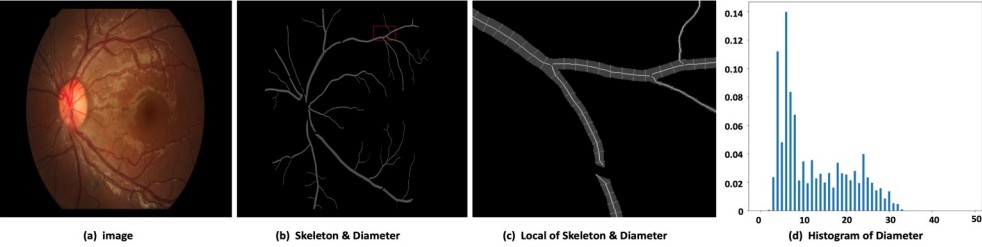

**Fig 2. Visualization of the retinal vessel diameters in the fundus.** (a) The raw 1880×2886 image. (b) Diameter of each point on the skeleton. c) Partially enlarged view of the red area in (b). (d) Histogram of the diameter map for each point on the skeleton (distance between pixels).

[23] bottleneck block (as shown in Fig 3(A)), we propose a Res2Net variant with better extraction capability of multi-scale features, with roughly the same computational cost. The 3×3 filter groups are replaced with smaller filter groups connected in a hierarchical residual-type manner. As shown in Fig 3(B), after the 1×1 convolution, the features are split into $k$ subsets, denoted by $x_i$, where $i \in \{1, 2, \ldots, k\}$. While all subsets have the same spatial size, the channel count for each subset is $1/k$ times that of the input feature map. Each subset $x_i$ (except for $x_1$) has a 3×3 convolution filter $F_i()$.

In SA module, we add the attention module to enforce the scale-attention capability. The details of proposed attention module are shown in Fig 4. Firstly, a max-pooling and an average-pooling are used to obtain the global information of each channel, which automatically highlight the relevant feature channels while suppressing irrelevant channels. Then, their results are summed and fed into a 1×1 convolutional layer followed by the sigmoid activation function. Finally, the output result is obtained by multiplying with the input.

## Scale-attention module

For the best possible transfer of the useful small-scale field-of-view features to large-scale features, we propose a scale-aware (SA) module, as shown in Fig 3(C). This SA module adds the attention model ($A_i()$) with the argument $y_i$, as shown in Fig 4. First, we get the attention map $A_i(y_i)$ and concatenate it with $x_{i+1}$, and then feed the result into $F_i()$. To decrease the

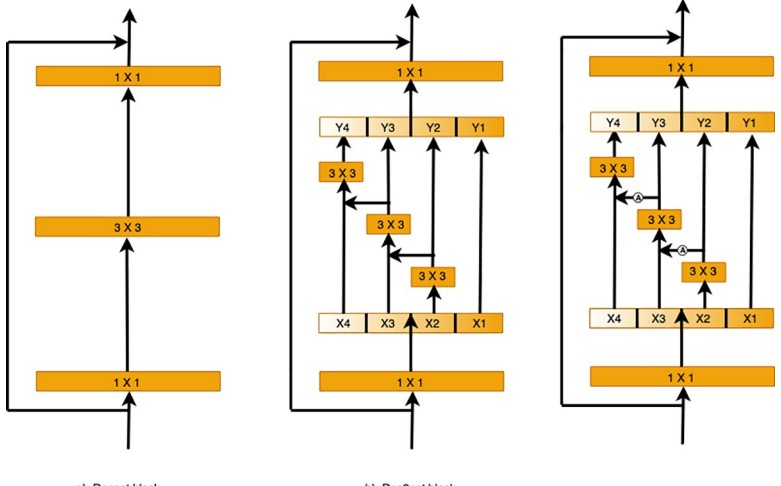

**Fig 3.** Comparing the Res2Net and ResNet blocks (with a scale dimension of k = 4): (a) The conventional building block in CNN variants. (b) Res2Net uses a group of 3×3 filters. (c) The SA module adds the attention module to enforce the scale-attention capability.

**Fig 4. Diagram of the attention module.** As illustrated, the attention module utilizes both max-pooling outputs and average-pooling outputs.

parameters while allowing the number of subsets $k$ to increase, we omit the 3×3 convolution for $x_1$ herein. Thus, $y_i$ can be written as:

$$
y_i = \begin{cases} x_i & i = 1; \\ F_i(x_i) & i = 2; \\ F_i((x_i;\ A_{i-1}(y_{i-1}))) & 2 < i \leq k. \end{cases}
$$

Each 3×3 convolutional operator $F_i()$ might get information from all feature subsets $\{x_j, j \leq i\}$. When a feature subset $x_j$ is processed by a 3×3 convolutional operator, the resulting output may have an enlarged receptive field compared to $x_j$. The high combinatorial complexity causes the SA module output to have different numbers and combinations of the receptive field size and scales.

We process the feature subsets in the SA module following a multi-scale approach, where local and global information is extracted. All subsets are concatenated and processed with 1×1 convolution, with the aim of achieving better fusion of information at different scales. More effective feature convolution is achieved by this split and concatenation strategy. With the objective of reducing the parameter count, a feature reuse approach is followed where the convolution for the first subset was omitted.

The scale dimension $k$ is used herein as a control parameter. A larger $k$ value might enable the learning of features with richer receptive field sizes, with insignificant concatenation-induced overheads in terms of computations and memory usage.

## The loss function

We employ an end-to-end deep-learning scheme as our underlying framework. Fig 1 asserts the need for training the proposed system to predict the segmentation label of each pixel. The loss is quantified by the commonly-used cross-entropy loss function.

$$
L_{ce} = -\frac{1}{n}\sum_{i=1}^{n}(y_i\, log\, (y'_i) + (1 - y_i)log\, (1 - y'_i))
$$

while the total loss is defined as:

$$
Loss = L_{ce} + \beta * \|W\|_2^2
$$

where $n$ denotes the number of pixels in the input image, $y'$ is the predicted output probability of a foreground pixel, and $y$ is the ground-truth pixel. We use $L_2$ regularization with a weight of $\beta = 0.0002$.

## Experimental evaluation and results

### Experimental settings

This section introduces the image preprocessing and data augmentation procedures for network training. And also provides ample details about our experimental setup.

Data transformations and augmentation are needed during training to avoid model overfitting. In medical imaging, an essential constraint of these transformations is that the output images must be quite realistic. To increase the training data variability while achieving this realism, we employ only two-dimensional rotations (through random angles) of each training batch. In particularly, we realize that the background color and illumination of different fundus images induce a large variability in pixel intensity. This variability is inherent in the training data. Contrast enhancement could be used to reduce this variability and increase the image quality in image preprocessing. And the impact of individual differences is avoided by using gray retinal images instead of color images [2, 25]. In addition, in this study we select the last epoch when the loss of the training model fluctuates less than 0.01 within 20 epochs as our final model for testing.

Our system was implemented using an Ubuntu 16.04 operating system, an Intel® Xeon® Gold 6148 CPU with a 2.40-GHz processor and 256-GB RAM, NVIDIA Tesla V100 GPU, a PyTorch backend, and cuDNN 9.0.

## Retinal vessel detection

For retinal vessel detection in fundus images, we evaluated our method on two publicly available datasets. Firstly, we used the 40-images DRIVE [26] dataset, for which two expert human annotations are available. The first annotation is typically used as the gold standard [4]. The DRIVE dataset consists of 20 training images and 20 testing images with the resolution of 584×565. Secondly, we examined the 28-images CHASE_DB1 database [27], with images from both eye sides for each of 14 children. A clear discrimination couldn't be reached between the healthy and diseased cases in CHASE_DB1. Thus, we adopted a scheme of stratified *k*-fold cross-validation, in which the input data is subdivided into equally-sized *k* folds, such that one fold is set for testing, while the other (*k-1*) folds are set for training. The results of *k* repetitions of this process are averaged to get pooled estimates of segmentation metrics. For performance consistency among cross-validation folds, the same settings and training initialization were used. For our CHASE_DB1 experiment, four folds were used with 7 images each divided almost evenly among the two eye sides. The ground-truth segmentation data for CHASE_DB1 was generated using Hoover's annotations. For the DRIVE images, binary segmentation masks are publicly available. Field-of-view (FOV) masks for CHASEE_DB1 were manually created following [28]. Example images and masks for this two datasets are demonstrated in Fig 5.

For blood vessel detection performance evaluation, several metrics were computed, namely sensitivity (SE), specificity (SP), accuracy (ACC), Matthews correlation coefficient (MCC), and the F1 score (F1) [4]. Moreover, for the receiver operating characteristic curve, we compute the area under the curve (AUC) and use it as well to assess the segmentation performance. For all of these metrics, a perfect detector gives a value of 1. A threshold value of 0.5 was

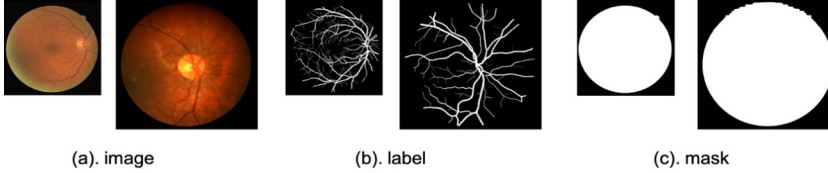

(a). image          (b). label          (c). mask

**Fig 5. Samples of the retinal fundus images and their H×W dimensions in pixels.** (a) Input images: Left: DRIVE (584×565); right: CHASE_DB1 (960×999). (b) Manual annotation of the retinal vessels. (c) Binary segmentation masks of the fundus images.

**Table 1. Segmentation performance metrics on two publically available retinal image datasets.**

| Datasets | | Method | MCC | SE | SP | ACC | AUC | F1 |
|---|---|---|---|---|---|---|---|---|
| DRIVE | Unsupervised | Zhao [29] | N/A | 0.7420 | 0.9820 | 0.9540 | 0.8620 | N/A |
| | | Azzopardi [30] | N/A | 0.7655 | 0.9704 | 0.9442 | 0.9614 | N/A |
| | | Roychowdhury [31] | N/A | 0.7395 | 0.9782 | 0.9494 | 0.8672 | N/A |
| | Supervised | U-Net [2] | N/A | 0.7537 | 0.9820 | 0.9531 | 0.9755 | 0.8142 |
| | | RU-Net [6] | N/A | 0.7792 | 0.9813 | 0.9556 | 0.9784 | 0.8171 |
| | | DE-Unet [25] | N/A | 0.7940 | 0.9816 | 0.9567 | 0.9772 | 0.8270 |
| | | SWT-UNet [32] | 0.8045 | 0.8039 | 0.9804 | 0.9576 | 0.9821 | 0.8281 |
| | | BTS-UNet [33] | 0.7923 | 0.7800 | 0.9806 | 0.9551 | 0.9796 | 0.8208 |
| | | Driu [34] | 0.7941 | 0.7855 | 0.9799 | 0.9552 | 0.9793 | 0.8220 |
| | | CS-Net [35] | N/A | 0.8170 | **0.9854** | **0.9632** | 0.9798 | N/A |
| | | **SA-Net (Ours)** | **0.8055** | **0.8252** | 0.9764 | 0.9569 | **0.9822** | **0.8289** |
| CHASE-DB1 | Unsupervised | Azzopardi [30] | N/A | 0.7585 | 0.9587 | 0.9387 | 0.9487 | N/A |
| | | Roychowdhury [31] | N/A | 0.7615 | 0.9575 | 0.9467 | 0.9623 | N/A |
| | Supervised | RU-Net [6] | N/A | 0.7756 | 0.9820 | 0.9634 | 0.9815 | 0.7928 |
| | | SWT-UNet [32] | 0.8011 | 0.7779 | **0.9864** | 0.9653 | 0.9855 | 0.8188 |
| | | BTS-UNet [33] | 0.7733 | 0.7888 | 0.9801 | 0.9627 | 0.9840 | 0.7983 |
| | | **SA-Net (Ours)** | **0.8102** | **0.8199** | 0.9827 | **0.9665** | **0.9865** | **0.8280** |

[a]N/A = Not Available, SWT-UNet shows the vessel segmentation results using fully convolutional neural networks, BTS-DSN gives the segmentation results with the multi-scale deeply-supervised networks with short connections.

applied to the probability maps to obtain the binary segmentation outputs. Only pixels inside the field of view were processed.

The performance of SA-Net was compared against relevant algorithms [29–31], in addition to recently-developed deep learning methods [2, 6, 25, 32–35]. Table 1 shows the comparative experimental results. Indeed, for DRIVE dataset, SA-Net achieves remarkable performance in terms of the F1, MCC, SE and AUC metrics, with values of 0.8289, 0.8055, 0.8252 and 0.9822, respectively. For CHASE_DB1, all of the MCC, SE, ACC, AUC and F1 metrics achieve excellent performance, with values of 0.8102, 0.8199, 0.9665, 0.9865 and 0.8280, respectively. For visualization results, example outputs are given in Fig 9. For the identification of the blood vessels at different scales, the SA-Net shows more continuous results and the detection of small blood vessels are especially close to the true labels. These results clearly indicate that devising the SA-Net architecture with multi-level scale-aware capabilities led to superior retinal vessel segmentation performance.

## Lung segmentation

In this section, we seek to segment the lung structures in 2D CT images. We evaluate SA-Net on 2D lung CT images provided by the Lung Nodule Analysis (LUNA) competition, in which two challenges have been made, namely nodule detection and false-positive reduction. The LUNA dataset contains 534 publically-available images of 512×512 pixels each, and corresponding segmentation masks [36]. We experiment with a cross-validation scheme in addition to a scheme with 80% training images and 20% testing ones, which same with the CE-Net [20]. Sample CT images are displayed in Fig 6.

The performance metrics used herein are the overlapping error $E$ (defined below), the accuracy and the sensitivity [20]. In addition to the average metric values are reported in Table 2.

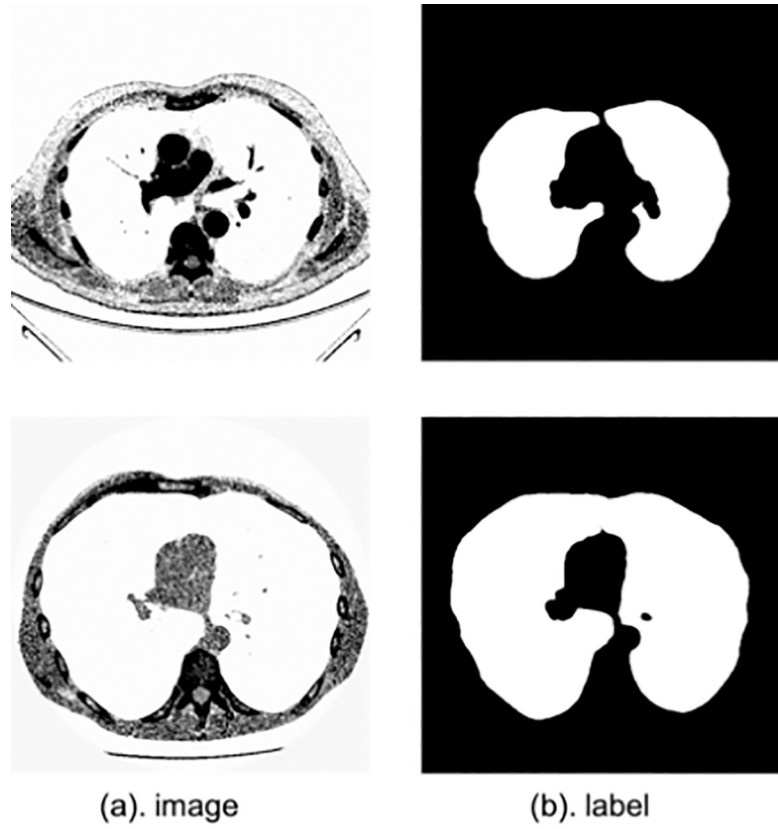

**Fig 6. Sample lung CT images with dimensions of H×W in pixels.**

The overlapping error is given by

$$E = 1 - \frac{Area(P \cap T)}{Area(P \cup T)}$$

where $P$ and $T$ denote the predicted and ground-truth lung segmentations, respectively.

From Table 2, as we can see that SA-Net reaches an overlapping error of 0.035, a sensitivity of 0.988, and an accuracy of 0.986. These values are better than the corresponding ones obtained by U-Net. Also, we compare SA-Net with the CE-Net [20] architecture, which pays more attention to high features. The overlapping error drops by 7.89% from 0.038 to 0.035, while the sensitivity increases from 0.980 to 0.988. Some examples are also given in Fig 9. As shown, because of the addition of the SA module, the lung structure can be better identified, and the segmentation result is more consistent than that of U-Net (which wrongly identifies non-lung tissues as lung ones). The results further emphasize the significance of our proposed SA blocks for lung segmentation.

## Artery/Vein classification

Our third application is A/V classification in fundus images. Motivated by the lack of a gold standard that allows objective comparison of approaches for this problem, Qureshi *et al.* [37] created a manually-annotated benchmark for A/V classification using the DRIVE dataset. The labeling was performed by one ophthalmologist and two computer vision scientists. A majority vote among the three experts was used to decide the ground-truth blood vessel class. Also, as

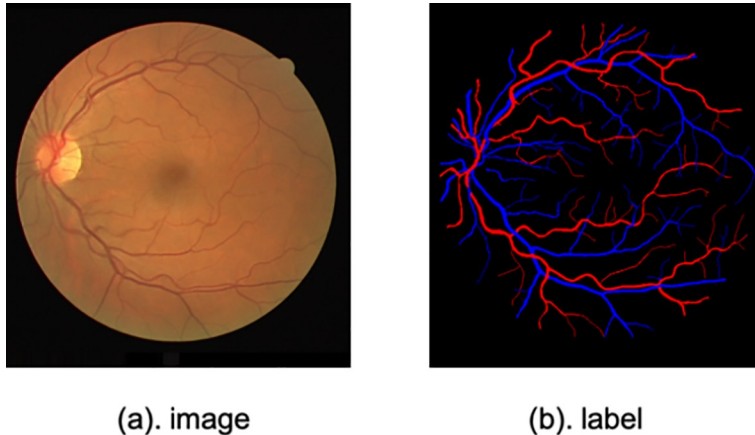

(a). image                    (b). label

**Fig 7. A sample retinal image from the DRIVE dataset and its corresponding artery/vein segmentation map.**

described by Hu *et al.* [38], ground-truth data was produced from the binary vessel segmentation created by the second expert. In this work, the ground-truth acts as a proxy for A/V classification by an independent expert. The training and testing subsets were obtained following the same schemes as those of the blood vessel detection problem, which means that 20 images are used for training and the remaining 20 are used for testing. The Balance accuracy (BACC), $SE_{AV}$, $SP_{AV}$, F1 score of arteries ($F1_A$) and F1 score of veins ($F1_V$) are used in this section to evaluate the SA-Net performance. The image and label samples are displayed in Fig 7.

$$SE_{AV} = \frac{TP}{TP + FN}$$

$$SP_{AV} = \frac{TN}{FP + TN}$$

$$Balance\ accuracy\ (BACC) = \frac{SE_{AV} + SP_{AV}}{2}$$

Where TP is the number of artery pixels correctly classified, TN is the number of vein pixels correctly classified, FP is the number of vein pixels mis-classified as artery pixels and FN is the number of artery pixels mis-classified as vein pixels.

In Table 3, the SA-Net result shows better performance. This demonstrates the validity of our model. As well, it can be seen that the results for veins are better than those for arteries. This difference is mainly because of the clinical manifestations, i.e., the color of the vein is dark red, while the arterial blood is light red. Some examples are given in Fig 9. We can see that the identification of blood vessels on different scales is better whether these vessels are veins or arteries. Because of the addition of the SA module, SA-Net works better in distinguishing arteries and veins, while the confusion between arteries and veins is serious for U-Net. These results reaffirm the superiority of the SA module in learning images with different scales.

## Blastocyst segmentation

The fourth application is blastocyst images segmentation. There is a detail introduction to the human embryo dataset in [40], which has been open sourced. This blastocyst dataset contains 235 blastocyst images collected from time-lapse. Various tissue labels are provided by the

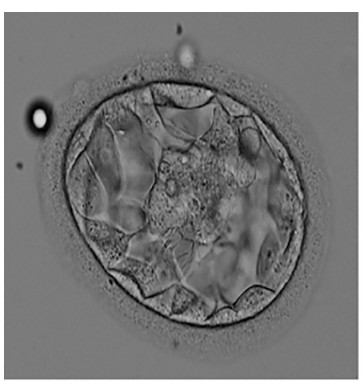
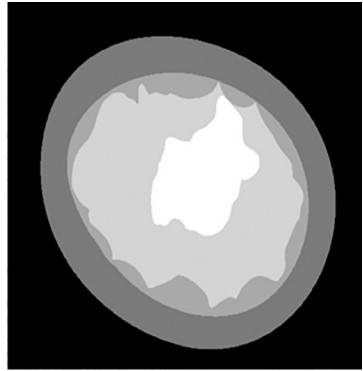

(a). image                           (b). label

**Fig 8. A sample blastocyst image from the blastocyst dataset and its manual label.**

Pacific Centre for Reproductive Medicine (PCRM). The training set accounts for 85% of all data while the test set accounts for the remaining 15%. And the division of the training set and test set is consistent with [41]. The sample is shown in Fig 8.

For blastocyst images segmentation performance evaluation, *Jaccard Index*, also called *Intersection-over-Union*, is utilized to evaluate segmentation performance. The metrics is same with the Blast-Net [41].

Our method is compared with several classic methods [9, 10, 41, 42]. Table 4 shows the comparative experimental results. As for Inner Cell Mass (ICM), Blastocoel, Trophectoderm (TE), Zona Pellucida (ZP) and Background segmentation, it can be seen that SA-Net have achieved significant improvement performance, which are 83.67%, 89.56%, 77.50%, 90.93% and 97.52%, respectively. Moreover, from Fig 9 we can see that SA-Net shows better segmentation performance compared with the basic U-Net. These results also have proved that the addition of the SA model has better comprehension for the multi-scale structural features of different organs.

## Ablation study

For further assessment of the SA module, we carried on an ablation study on the DRIVE dataset. The U-Net architecture was used as the baseline model. In one variant, we replaced the middle skip connections with the bottleneck and ResNet, and with the bottleneck and the SA module for Res2Net. We then verified the importance of the multiscale and attentional features.

In Table 5, adding the residual module increases the ACC, AUC and F1 scores to 0.9566, 0.9817 and 0.8244, respectively. We can see that the residual module is indeed very helpful for improving the learning characteristics. When adding the multi-scale features to the Res2Net

**Table 2. Segmentation performance measures for lung image datasets.**

| Method | E | ACC | SE |
|---|---|---|---|
| U-Net [2] | 0.087 | 0.975 | 0.938 |
| CE-Net [20] | 0.038 | **0.990** | 0.980 |
| SA-Net (Ours) | **0.035** | 0.986 | **0.988** |

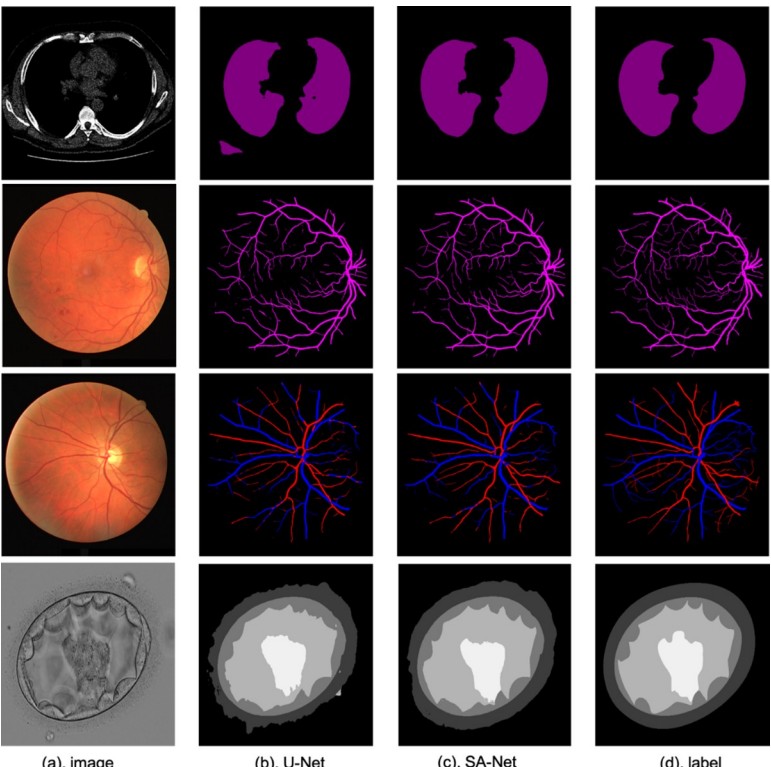

(a). image    (b). U-Net    (c). SA-Net    (d). label

**Fig 9. Example results for lung segmentation, detection of retinal blood vessels, artery/vein classification and blastocyst segmentation.** From top to bottom: lung segmentation, retinal vessel detection, artery/vein classification and blastocyst segmentation.

learning, it can be found that the sensitivity and F1 scores had a significant increase, while the other indicators showed no obvious improvements, and even the AUC metric slightly decreased. After the introduction of the multi-scale scheme, the model ability to understand the scale features became stronger and the sensitivity became higher. However, some useless information was inadvertently introduced, resulting in feature redundancy. When we added our proposed SA module, we can see that all indicators except for the specificity have increased, among which MCC, SE, ACC, AUC and F1 scores increase to 0.8055, 0.8252, 0.9569, 0.9822, and 0.8289, respectively. These results demonstrate the effectiveness of the proposed scale-attention module to a certain extent.

## Discussion and conclusions

The task of segmenting different tissue types in medical imaging is very critical, and good segmentation results are conducive to the analysis of some tasks in later stages. In this paper, we proposed an end-to-end scale-attention deep learning network, and experiment results show that SA-Net has achieved superior performance results in retinal vessel detection, lung

**Table 3. Performances of different A/V classification methods on DRIVE dataset.**

| Method | BACC | $SE_{AV}$ | $SP_{AV}$ | $F1_A$ | F1V |
|---|---|---|---|---|---|
| U-Net [2] | 0.9122 | 0.9145 | 0.9083 | 0.7089 | 0.7586 |
| DoS [39] | N/A | 0.9190 | 0.9150 | N/A | N/A |
| SA-Net (Ours) | **0.9351** | **0.9345** | **0.9347** | **0.7336** | **0.7802** |

**Table 4. Blastocyst segmentation performance measures on the blastocyst dataset.**

| Method | Mean | Background | Blastocoel | ZP | TE | ICM |
|---|---|---|---|---|---|---|
| U-Net [2] | 0.8137 | 0.9404 | 0.7941 | 0.7932 | 0.7506 | 0.7903 |
| PSPNet [9] | 0.8151 | 0.9460 | 0.7926 | 0.8057 | 0.7483 | 0.7828 |
| DeepLab V3 [21] | 0.8165 | 0.9449 | 0.7835 | 0.8084 | 0.7398 | 0.8060 |
| Blast-Net [41] | 0.8285 | 0.9474 | 0.8079 | 0.8115 | 0.7652 | 0.8107 |
| TernausNet [42] | 0.8142 | 0.9450 | 0.7861 | 0.8024 | 0.7616 | 0.7758 |
| **SA-Net (Ours)** | **0.8783** | **0.9752** | **0.8956** | **0.9093** | **0.7750** | **0.8367** |

**Table 5. Ablation study segmentation results on the DRIVE dataset.**

| Method | MCC | SE | SP | ACC | AUC | F1 | Params |
|---|---|---|---|---|---|---|---|
| Backbone [2] | N/A | 0.7537 | **0.9820** | 0.9531 | 0.9755 | 0.8142 | 34.5M |
| Backbone+ResNet | 0.8013 | 0.8039 | 0.9793 | 0.9566 | 0.9817 | 0.8244 | 49.5M |
| Backbone+Res2Net | 0.8026 | 0.8148 | 0.9775 | 0.9566 | 0.9816 | 0.8260 | 161M |
| Backbone+SA | **0.8055** | **0.8252** | 0.9764 | **0.9569** | **0.9822** | **0.8289** | 194M |

segmentation, artery/vein classification and blastocyst segmentation tasks. More importantly, it can be seen from the visualization results in Fig 9 that SA-Net shows better performance for segmentation of different medical images with different scales tissues compared with traditional U-Net architecture, indicating that SA-Net can better learn the features at different scales. In this work, the proposed SA-Net achieves effective segmentation in various medical images, it can be considered as a better model for 2D small-sample medical image segmentation. While we only explored 2D medical images in this work, we will explore the application of SA-Net for segmenting 3D medical images and consider reducing the computational cost of the model in the future.

## Author Contributions

**Conceptualization:** Hua Wang.

**Formal analysis:** Hua Wang, Jie Wang.

**Funding acquisition:** Jicong Zhang.

**Investigation:** Jingfei Hu, Hua Wang, Jie Wang, Yunqi Wang, Fang He.

**Methodology:** Jingfei Hu, Hua Wang.

**Project administration:** Jicong Zhang.

**Resources:** Jicong Zhang.

**Supervision:** Jicong Zhang.

**Visualization:** Jingfei Hu, Hua Wang.

**Writing – original draft:** Jingfei Hu.

**Writing – review & editing:** Hua Wang.

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
