## [Decision Letter · Decision Letter 0]

17 Nov 2020

PONE-D-20-30729

SA-Net: A Scale-Attention Network for Medical Image Segmentation

PLOS ONE

Dear Dr. Zhang,

Thank you for submitting your manuscript to PLOS ONE. After careful consideration, we feel that it has merit but does not fully meet PLOS ONE’s publication criteria as it currently stands. Therefore, we invite you to submit a revised version of the manuscript that addresses the points raised during the review process.

We look forward to receiving your revised manuscript.

Kind regards,

Yuankai Huo, Ph.D.

Academic Editor

PLOS ONE

"This research was supported by Hefei Innovation Research Institute, Beihang University and ‘the Thousand Talents Plan’ Workstation between Beihang University and Jiangsu Yuwell Medical Equipment and Supply Co. Ltd. The authors are grateful to all study participants."

"This work was supported by the National Key Research and Development Program of China under Grant 2016YFF0201002, the National Natural Science Foundation of China under Grant 61572055, the University Synergy Innovation Program of Anhui Province GXXT-2019-044. These awards were received by Jicong Zhang. YES - Specify the role(s) played."

Additionally, because some of your funding information pertains to commercial funding, we ask you to provide an updated Competing Interests statement, declaring all sources of commercial funding.

In your Competing Interests statement, please confirm that your commercial funding does not alter your adherence to PLOS ONE Editorial policies and criteria by including the following statement: "This does not alter our adherence to PLOS ONE policies on sharing data and materials.” as detailed online in our guide for authors  http://journals.plos.org/plosone/s/competing-interests.  If this statement is not true and your adherence to PLOS policies on sharing data and materials is altered, please explain how.

Please include the updated Competing Interests Statement and Funding Statement in your cover letter. We will change the online submission form on your behalf.

Reviewers' comments:

Reviewer's Responses to Questions

**Comments to the Author**

1. Is the manuscript technically sound, and do the data support the conclusions?

Reviewer #1: Yes

Reviewer #2: Yes

2. Has the statistical analysis been performed appropriately and rigorously? 

Reviewer #1: Yes

Reviewer #2: Yes

3. Have the authors made all data underlying the findings in their manuscript fully available?

Reviewer #1: Yes

Reviewer #2: Yes

4. Is the manuscript presented in an intelligible fashion and written in standard English?

Reviewer #1: Yes

Reviewer #2: Yes

5. Review Comments to the Author

Reviewer #1: This paper proposed a scale-attention network upon the backbone network of UNet. And designed multiple attention modulus at different scales, on segmentation tasks. The attention mechanism at scales improved the extraction of global-local features. The proposed method is trained and tested on multiple datasets including two public retina vessel datasets and the LUNA. The method is also validated on the artery/vein classification problem and blastocyst segmentation. The study across multiple datasets are impressive. In addition, the method is compared with extensive baseline methods in Table 1, which is a bonus and provides a good reference for future studies.

Overall, this is a well evaluated paper with multiple applications and large-scale experiments. However, I have several concerns about several places, especially in the description of abstract/introduction and discussion. It would be great to further tune some statements, please see as follows:

1. The abstract address that “uses an attention module to learn and understand which features are the most important for medical image segmentation”. However, this paper didn’t evaluate the feature importance or have the conclusion on which features are the most important. I would recommend the authors to add the evaluation of different features effects or to rephrase the sentence.

2. I suggest changing the description of “not only a waste of manpower and time but also prone” in the first sentence of introduction to “manual effort is time-consuming and tedious”, we should be humble to previous manual work by radiologists and clinicians, these are hard tasks and defines many critical medical problems.

3. “In particular, U-Net [2] achieved the best segmentation accuracy for neuronal structures in electron microscopy.” This claim lacks a citation, could the authors add the citation on the best performed paper of “neuronal structures in electron microscopy”?\\

4. Suggestion to reduce redundancy: “and its performance does not deteriorate even when large enough datasets are lacking” to “and its performance does not deteriorate at low data regime”, many other places are similar, it will be great if the authors could further read and try reduce redundancy.

5. The fourth paragraph in Introduction, “Nevertheless, …., these variants still rely on cascaded multi-stage CNNs. Therefore, we emphasize the design of particular good multi-scale features”. This sentence is confusing, I believe the propose method is still based on encoder-decoder architecture and used the cascaded multi-stage CNNs. Could the authors rephrase the sentence and highlight the difference between the proposed work and convention UNet?

6. The legend of Figure 1 and Figure 2 are too small to see, could enlarge the legend.

7. The experiment setting separated the datasets into training and testing or cross-validation, since there is no validation set, how the final model is selected for testing?

8. Followed by the first concern, the authors highlighted the ability to “better learn and understand the features at different scales” at the Discussion and Conclusion, it would be more intuitive to visualize the attention maps at different scales for reader, and discuss the difference with non-multi-scale attention. Otherwise, I would suggest removing this statement.

Summary:

This is a well-evaluated paper. The large-scale experiments and comparisons are impressive. The application on five different datasets is a bonus. Overall, this is a great study, it should be of potential interest of readers if authors could address above issues and further fine-tune some descriptions.

Reviewer #2: Overall, this work proposes a scale-attention module to the Res2Net based UNet for the task of classification and segmentation in the medical imaging field. Experiments are done on 4 publicly available datasets and with one ablation study.

Cons & Questions:

1) The claimed contribution is the proposed scale-attention module. However, according to the ablation study in Table V, the improvement of SA compared with Res2Net is really marginal, only 0.03% in ACC, 0.06% in AUC and 0.29% in F1, which could also be in the std range if running different algorithms for multiple times. Also, the analysis in the main text "In fact, the MCC, SE, ACC, AUC and F1 scores increase by 0.36%, 1.28%, 0.03%, 0.06%, and 0.35%, respectively" is not consistent with the numbers in Table V. Can authors check that?

2) What is the computation cost comparison for the ablation study in Table V? I can image a marginal improvement can be obtained by a bit more computation cost.

3) Some important details of some datasets are missing. The training and testing split of the DRIVE dataset and the A/V classification dataset are missing. What is the GPU version you are using for the training and testing?

4) In table IV, the reference number of Blast-Net is mixed to be the TernausNet paper.

Pros:

1) The written is clear. The scale problem is a pain point in the medical imaging, even for the computer vision.

2) The experiments are done on many datasets.

6. PLOS authors have the option to publish the peer review history of their article (what does this mean?). If published, this will include your full peer review and any attached files.

Reviewer #1: No

Reviewer #2: No

---

## [Author Response · Author response to Decision Letter 0]

24 Dec 2020

Dear Editors and Reviewers:

We thank you very much for giving us an opportunity to revise our manuscript. Thanks a lot for the helpful comments and recommends and thank you for your time spent. According to the reviewers' comments, we have revised the manuscript. Revised portion are marked in red in the paper. 

The main corrections in the paper and the responds to the comments are as follows: 

Reviewer #1: 

General comments

This paper proposed a scale-attention network upon the backbone network of UNet. And designed multiple attention modulus at different scales, on segmentation tasks. The attention mechanism at scales improved the extraction of global-local features. The proposed method is trained and tested on multiple datasets including two public retina vessel datasets and the LUNA. The method is also validated on the artery/vein classification problem and blastocyst segmentation. The study across multiple datasets are impressive. In addition, the method is compared with extensive baseline methods in Table 1, which is a bonus and provides a good reference for future studies.

Overall, this is a well evaluated paper with multiple applications and large-scale experiments. However, I have several concerns about several places, especially in the description of abstract/introduction and discussion. It would be great to further tune some statements, please see as follows:

1. The abstract address that “uses an attention module to learn and understand which features are the most important for medical image segmentation”. However, this paper didn’t evaluate the feature importance or have the conclusion on which features are the most important. I would recommend the authors to add the evaluation of different features effects or to rephrase the sentence. Response: Thanks for your suggestion. Indeed, our description is confusing. To avoid confusion, we have rephrased the sentence “uses an attention module to learn and understand which features are the most important for medical image segmentation” in the abstract as “uses an attention module to enforce the scale-attention capability. SA-Net can better learn the multi-scale features and achieve more accurate segmentation for different medical image.” according to the reviewer’s comment.

2. I suggest changing the description of “not only a waste of manpower and time but also prone” in the first sentence of introduction to “manual effort is time-consuming and tedious”, we should be humble to previous manual work by radiologists and clinicians, these are hard tasks and defines many critical medical problems. Response: Thanks for the suggestion. We think the reviewer’s comment is very reasonable and we are also very ashamed of the inappropriate expression. Our description is indeed inappropriate, we have revised the description of “not only a waste of manpower and time but also prone” in the first sentence of introduction according to the reviewer’s suggestion. As shown in the first sentence of Introduction marked in red.

3. “In particular, U-Net [2] achieved the best segmentation accuracy for neuronal structures in electron microscopy.” This claim lacks a citation, could the authors add the citation on the best performed paper of “neuronal structures in electron microscopy”? Response: Thanks for pointing it out. We are very sorry that we ignored this citation, in the revised version, we have added reference [3]：“Ibtehaz N, Rahman M S. MultiResUNet: Rethinking the U-Net architecture for multimodal biomedical image segmentation[J]. Neural Networks, 2020, 121: 74-87.”, which is a reference on the best performed paper of “neuronal structures in electron microscopy”. The revised version is “U-Net [2] and U-Net variants models have been successfully used in segmenting biomedical images of neuronal structures. In particular, MultiResUNet [3] achieved the best segmentation accuracy for neuronal structures in electron microscopy.”

4. Suggestion to reduce redundancy: “and its performance does not deteriorate even when large enough datasets are lacking” to “and its performance does not deteriorate at low data regime”, many other places are similar, it will be great if the authors could further read and try reduce redundancy.

Response: Thanks for the suggestion. In order to reduce redundancy, we have carefully read and revised the whole manuscript according to the reviewer’s comment. The revised place has been marked in red in the manuscript.

5. The fourth paragraph in Introduction, “Nevertheless, …., these variants still rely on cascaded multi-stage CNNs. Therefore, we emphasize the design of particular good multi-scale features”. This sentence is confusing, I believe the propose method is still based on encoder-decoder architecture and used the cascaded multi-stage CNNs. Could the authors rephrase the sentence and highlight the difference between the proposed work and convention UNet?

Response: Thanks for the suggestion. We are very sorry that improper expression of this sentence makes you feel confused. In order to improve the quality of the manuscript, we have rephrased the sentence according to the reviewer’s comment as “U-Net and U-Net like models has been showing impressive potential in segmenting medical images, but the performance of these models will be poor when the target organ exhibits large shape and size variations among patients. Therefore, design good multi-scale features for medical images segmentation is essential. However, creating multi-scale representations requires feature extractors to use receptive fields of considerable variations to give a detailed account of parts, objects, or context at all possible scales. The natural way for CNNs to extract coarse-to-fine multi-scale features is to utilize a convolutional operator stack. Such inherent CNN capability of extracting multi-scale features leads to good representations for handling numerous medical image analysis tasks”. We hope this revision will satisfy you.

6. The legend of Figure 1 and Figure 2 are too small to see, could enlarge the legend.

Response: Thanks for pointing it out. We have replaced a new figure with enlarged the legend according to the reviewer’s suggestion. As shown in Fig1 and Fig 2.

7. The experiment setting separated the datasets into training and testing or cross-validation, since there is no validation set, how the final model is selected for testing?

Response: Thanks for the question. When the loss of the training model continuously fluctuates less than 0.01 within 20 epochs, the training is stopped, and then the last epoch is selected as the final model for testing. In order to make the manuscript more clear and easier for readers to understand, we have also made corresponding explanations in the last sentence of the second paragraph of the Experimental settings section, which is described as: “In addition, in this study we select the last epoch when the loss of the training model fluctuates less than 0.01 within 20 epochs as our final model for testing.”

8. Followed by the first concern, the authors highlighted the ability to “better learn and understand the features at different scales” at the Discussion and Conclusion, it would be more intuitive to visualize the attention maps at different scales for reader, and discuss the difference with non-multi-scale attention. Otherwise, I would suggest removing this statement.

Response: Thanks for your suggestion. In this paper, the Fig 9 is a visualization of the segmentation results for different medical images, in which lung CT images, retinal blood vessel images and blastocyst images have different resolutions, and different tissues also have different scales. It can be clearly seen from Fig 9 that the segmentation results of SA-Net are better than the traditional U-Net architecture. For example, whether it is a small-scale retinal blood vessel or a large-scale lung image, SA-Net shows better performance. Moreover, the Inner Cell Mass, Blastocoel, Trophectoderm, Zona Pellucida and Background have different scales in blastocyst image, but the SA-Net performance is still better compared with U-Net without multi-scale attention module. Therefore, these experimental results make us think that our proposed model can better learn the features at different scales. It may be that our expression is not clear enough, which makes you feel puzzled. We are very sorry about this, in response to this problem, we have revised this sentence in the manuscript and hope to make the expression of this sentence more clearly. The revised expression is shown in the red sentence in the Discussion and Conclusion section.

Summary:

This is a well-evaluated paper. The large-scale experiments and comparisons are impressive. The application on five different datasets is a bonus. Overall, this is a great study, it should be of potential interest of readers if authors could address above issues and further fine-tune some descriptions.

Reviewer #2: 

General comments Overall, this work proposes a scale-attention module to the Res2Net based UNet for the task of classification and segmentation in the medical imaging field. Experiments are done on 4 publicly available datasets and with one ablation study.

1) The claimed contribution is the proposed scale-attention module. However, according to the ablation study in Table V, the improvement of SA compared with Res2Net is really marginal, only 0.03% in ACC, 0.06% in AUC and 0.29% in F1, which could also be in the std range if running different algorithms for multiple times. Also, the analysis in the main text "In fact, the MCC, SE, ACC, AUC and F1 scores increase by 0.36%, 1.28%, 0.03%, 0.06%, and 0.35%, respectively" is not consistent with the numbers in Table V. Can authors check that?

Response: Many thanks for your suggestions and comments. We are very sorry that we didn’t explain clearly how the increase rate is calculated, which makes you confused. We calculate the increase percentage of MCC, SE, ACC, AUC and F1 as follows: , however, we know that you use the following formula to calculate the increase percentage: , so our results are not consistent with yours. If calculate increase percentage according to our formula, our result is correct. But it is true that our calculation method is prone to confusion, so we have revised the manuscript according to your calculation method. The revised version is consistent with your results. 

In addition, comparing with Res2Net, indeed the improvement of SA is really marginal in terms of ACC, AUC and F1 indicators, but the improvement of sensitivity is significant, from 0.8148 to 0.8252 and the increase percentage is 1.04%. The sensitivity is a more important evaluation metrics compared to other metrics in medical image analysis. Therefore, the results of ablation experiments in Table 5 can demonstrate the importance of the proposed scale-attention module to a certain extent. In order to make the manuscript more clear and easier for readers to understand, we have revised the last few sentences of the Ablation study section as “When we added our proposed SA module, we can see that all indicators except for the specificity have increased, among which MCC, SE, ACC, AUC and F1 scores increase by 0.29%, 1.04%, 0.03%, 0.06%, and 0.29%, respectively. Particularly, the improvement of sensitivity is significant, these results can demonstrate the importance of the proposed scale-attention module to a certain extent.”

The revised sentence is:“When we added our proposed SA module, we can see that all indicators except for the specificity have increased, among which MCC, SE, ACC, AUC and F1 scores increase by 0.29%, 1.04%, 0.03%, 0.06%, and 0.29%, respectively. Particularly, the improvement of sensitivity is significant, so it can demonstrate the importance of the proposed scale-attention module to a certain extent.”

 2) What is the computation cost comparison for the ablation study in Table V? I can image a marginal improvement can be obtained by a bit more computation cost. 

Response: Thanks for the comments. We have added the computation cost in the last column of Table 5. As we can see from Table 5, it is indeed as you image that the computational cost is increasing with the improvement of performance. We are considering how to reduce the computational cost without reducing the performance of the model, this will also be one of our future research work.

3) Some important details of some datasets are missing. The training and testing split of the DRIVE dataset and the A/V classification dataset are missing. What is the GPU version you are using for the training and testing? Response: Many thanks for your suggestions. a) For the first question, we have added the details of the training and testing split of the DRIVE dataset in the fourth sentence of the first paragraph of the Retinal vessel detection section, and also added the details of the training and testing split of the A/V classification dataset in the third sentence at the end of the first paragraph of the Artery/vein classification section. The description in the manuscript is “The DRIVE dataset consists of 20 training images and 20 testing images with the resolution of 584×565.”and “The training and testing subsets were obtained following the same schemes as those of the blood vessel detection problem, which means that 20 images are used for training and the remaining 20 are used for testing.”, respectively. We have marked red for these modifications in our manuscript. b) For the second question, the GPU version we are using for the training and testing is NVIDIA Tesla V100 GPU. We are very sorry that we missed this in the manuscript, we have added the GPU version in the Experimental settings section, as shown in the last sentence of this section.

4) In table IV, the reference number of Blast-Net is mixed to be the TernausNet paper. Response: Thanks for pointing it out. We have changed the order of previous references [40] and [41] in the section of References. It should be noted that since we have added a new reference to the manuscript according to the reviewer1’s comments, the previous references [40] and [41] have become [41] and [42].

Pros:

1) The written is clear. The scale problem is a pain point in the medical imaging, even for the computer vision.

2) The experiments are done on many datasets.

---

## [Decision Letter · Decision Letter 1]

1 Feb 2021

PONE-D-20-30729R1

SA-Net: A Scale-Attention Network for Medical Image Segmentation

PLOS ONE

Dear Dr. Zhang,

Thank you for submitting your manuscript to PLOS ONE. After careful consideration, we think the paper is acceptable after the final minor correction. Therefore, we invite you to submit a revised version of the manuscript that addresses the points soon to get your paper published. 

We look forward to receiving your revised manuscript.

Kind regards,

Yuankai Huo, Ph.D.

Academic Editor

PLOS ONE

Additional Editor Comments (if provided):

This is paper is considered to be accepted after the final minor correction. Please make sure that the minor issues from the reviewer 1 is corrected.

"The emphasis of "improvement of sensitivity is significant" in the Ablation is not proper. First, the 1% improvement cannot be claimed as a significant one. Second, the comparison of sensitivity only is not fair since the specificity is worse. So I would suggest a rephrase. Apart from this, my other comments are addressed."

Reviewers' comments:

Reviewer's Responses to Questions

**Comments to the Author**

1. If the authors have adequately addressed your comments raised in a previous round of review and you feel that this manuscript is now acceptable for publication, you may indicate that here to bypass the “Comments to the Author” section, enter your conflict of interest statement in the “Confidential to Editor” section, and submit your "Accept" recommendation.

Reviewer #1: All comments have been addressed

Reviewer #2: All comments have been addressed

2. Is the manuscript technically sound, and do the data support the conclusions?

Reviewer #1: Yes

Reviewer #2: Yes

3. Has the statistical analysis been performed appropriately and rigorously? 

Reviewer #1: Yes

Reviewer #2: Yes

4. Have the authors made all data underlying the findings in their manuscript fully available?

Reviewer #1: Yes

Reviewer #2: (No Response)

5. Is the manuscript presented in an intelligible fashion and written in standard English?

Reviewer #1: Yes

Reviewer #2: Yes

6. Review Comments to the Author

Reviewer #1: The authors did very well in addressing the comments. The revisions significantly clarify the concerns of the paper by including statements of their work. It is now clear that the author's approach is novel, and statements are clear.

Reviewer #2: The emphasis of "improvement of sensitivity is significant" in the Ablation is not proper. First, the 1% improvement cannot be claimed as a significant one. Second, the comparison of sensitivity only is not fair since the specificity is worse. So I would suggest a rephrase. Apart from this, my other comments are addressed.

All in all, although I still think the added SA structure based on Res2Net is a marginal improvement at the cost of extra parameters (params used from backbone to the backbone+SA: 34M >> 49M >> 161M >> 194M), this manuscript is clear-written with extensive valuable experiments, which would be good for readers to know. So I give an accept recommendation.

7. PLOS authors have the option to publish the peer review history of their article (what does this mean?). If published, this will include your full peer review and any attached files.

Reviewer #1: **Yes: **Yucheng Tang

Reviewer #2: No

---

## [Author Response · Author response to Decision Letter 1]

3 Feb 2021

Dear Editors and Reviewers:

We thank you very much for giving us an opportunity to revise our manuscript again. Thanks a lot for the helpful comments and recommends and thank you for your time spent. According to the reviewers' comments, we have revised the manuscript. Revised portion are marked in red in the paper. 

The main corrections in the paper and the responds to the comments are as follows: 

Reviewer #2: The emphasis of "improvement of sensitivity is significant" in the Ablation is not proper. First, the 1% improvement cannot be claimed as a significant one. Second, the comparison of sensitivity only is not fair since the specificity is worse. So I would suggest a rephrase. Apart from this, my other comments are addressed.

Response: Thanks for the suggestion. We think this comment is very reasonable and we have rephrased the sentence. In fact, the 1% improvement is slightly, so in order not to confuse the reader, we directly replaced the percentage of increase with values. And in manuscript we rephrase the sentence as:“When we added our proposed SA module, we can see that all indicators except for the specificity have increased, among which MCC, SE, ACC, AUC and F1 scores increase to 0.8055, 0.8252, 0.9569, 0.9822, and 0.8289, respectively. These results demonstrate the effectiveness of the proposed scale-attention module to a certain extent.”

---

## [Editor Report · Decision Letter 2]

8 Feb 2021

SA-Net: A Scale-attention network for medical image segmentation

PONE-D-20-30729R2

Dear Dr. Zhang,

We’re pleased to inform you that your manuscript has been judged scientifically suitable for publication and will be formally accepted for publication once it meets all outstanding technical requirements.

Kind regards,

Yuankai Huo, Ph.D.

Academic Editor

PLOS ONE
---

## [Editor Report · Acceptance letter]

5 Apr 2021

PONE-D-20-30729R2 

SA-Net: A Scale-attention network for medical image segmentation 

Dear Dr. Zhang:

I'm pleased to inform you that your manuscript has been deemed suitable for publication in PLOS ONE. Congratulations! Your manuscript is now with our production department. 

Kind regards, 

on behalf of

Dr. Yuankai Huo 

Academic Editor

PLOS ONE